# Cervical Assessment for Predicting Preterm Birth—Cervical Length and Beyond

**DOI:** 10.3390/jcm10040627

**Published:** 2021-02-07

**Authors:** Lee Reicher, Yuval Fouks, Yariv Yogev

**Affiliations:** 1Tel Aviv Sourasky Medical Center, Lis Hospital for Women, 6 Weizman Street, Tel-Aviv 6423906, Israel; fouksi@gmail.com (Y.F.); yarivy@tlvmc.gov.il (Y.Y.); 2Sackler Faculty of Medicine, Tel Aviv University, Tel-Aviv 6423927, Israel

**Keywords:** short cervix, cervical insufficiency, spontaneous preterm birth, transvaginal ultrasound, preterm birth

## Abstract

Preterm birth is considered one of the main etiologies of neonatal death, as well as short- and long-term disability worldwide. A number of pathophysiological processes take place in the final unifying factor of cervical modifications that leads to preterm birth. In women at high risk for preterm birth, cervical assessment is commonly used for prediction and further risk stratification. This review outlines the rationale for cervical length screening for preterm birth prediction in different clinical settings within existing and evolving new technologies to assess cervical remodeling.

## 1. Introduction

Preterm birth (PTB), defined as delivery that occurs between 20 and 37 weeks of gestation, is a major obstetric and global health concern with a rate of 5–18% of pregnancies worldwide [1]. It is the largest direct cause of neonatal mortality and may be associated with serious morbidity in the surviving infants [2]. PTB is the final common element for the involvement of a variety of factors. All mechanisms for PTB alignment with the final unifying process of cervical remodeling [3]. According to this rationale, a cervical assessment may potentially detect preliminary pathological changes prior to the onset of overt symptoms and signs of PTB. This may be beneficial for preventive measures of PTB.

This review explores both the cervical assessment tests that are currently used in daily clinical practice, as well as novel emerging techniques of cervical assessment for predicting PTB, mainly in asymptomatic women.

## 2. The Preterm Parturition Syndrome

In 2006, Romero et al. described the preterm parturition syndrome, as a heterogeneous condition with preterm labor as the final common endpoint [3]. They proposed that spontaneous PTB (sPTB) results from mixed pathological activation of one or more of the signals that subsequently initiate spontaneous preterm labor. The pathological processes involved in preterm parturition syndrome fall into the broad categories of anatomical, physiological, biochemical, endocrinological, immunological, and clinical events that occur in the mother and/or fetus [4]. Although cervical insufficiency can result in cervical shortening and sPTB, this multifactorial model implies that shortening of the cervix may often be simply an early sign that accompanies some of the other abnormal processes which lead to a PTB.

## 3. The Role of Cervical Length in Predicting sPTB

One of the early changes that precede sPTB is cervical shortening, which may be detected even several weeks prior to the onset of active labor [5]. Currently, mid-trimester cervical length (CL) assessment by transvaginal ultrasound (TVUS) is one of the most commonly used tools for the prediction sPTB [6,7]. Regardless of obstetrical history, the risk of sPTB is inversely proportional to cervical length [8], while women with both a history of a PTB and a short cervix being at the highest risk [9]. However, as research addressing this topic is heterogenic with regards to the population that was studied [7] (e.g., women with prior preterm birth, women with prior cervical surgery, multiple gestations) and the clinical scenarios in which CL was used for predicting PTB (e.g., various gestational age at testing, symptomatic vs. asymptomatic women), it is important to address those aspects to optimize the use of this important tool.

## 4. What Is the Optimal Gestational Age for Cervical Length Screening?

Most clinical guidelines addressing this issue recommend performing cervical length screening between 16–24 weeks of gestation for asymptomatic women with a history of PTB [10,11]. It should not be routinely measured before 16 weeks of gestation [12] as the predictive accuracy of first and early second-trimester CL assessment for PTB is low, especially in asymptomatic women without an history of PTB [13,14,15]. Moreover, routine CL screening is not recommended after 24 weeks of gestation in asymptomatic women, given that most studies exploring various interventions aimed for prevention of PTB (e.g., cervical cerclage, vaginal progesterone) have most frequently used 24 weeks of gestation as the maximum bound for screening and initiation of therapies or interventions [16,17,18]. Nevertheless, it should be noted that asymptomatic women might still benefit from early identification of a short cervix, including transfer to a tertiary center, admission to a high-risk ward, and administration of corticosteroids and magnesium sulfate.

Of note, the diagnosis of a short cervix is generally limited to pregnant women, as CL measurements performed in non-pregnant women are not useful for predicting sPTB [19,20].

## 5. The Preferred Approach of Cervical Length Measurement

There are three common methods for sonographic cervical assessment: TVUS, transabdominal (TAUS), and transperineal (TPUS, also called translabial). CL measured by TVUS is associated with better prediction of a PTB than other approaches, and it is, therefore considered the gold standard [21,22,23] for CL measurement. In contrast to the transabdominal method, transvaginal cervical ultrasonography is less influenced by maternal obesity, cervix position, and shadowing from the fetal presenting part [24,25].

Previous studies reported that the sensitivity of TAUS to identify a TVUS-confirmed short cervix (i.e., <25 mm) ranges from 44.7% (using a TAUS cutoff of 25 mm) to 96.1% (using a TAUS cutoff of 36 mm) [26,27]. It is also worth noting that all randomized trials suggesting the effectiveness of treatment of women with a short cervix have been used by TVUS to assess cervical length [16,17,28].

### 5.1. What Cervical Length Threshold Should Be Used for Prediction of sPTL?

The predictive accuracy of a short CL for predicting sPTB is primarily related to the cutoff used. The reported sensitivity of a CL ≤ 25 mm for a PTB among high and low risk women varies from 6% to 76% [29]. One of the hallmark studies addressing the association of CL and the risk of PTB was conducted by Iams et al. [5]. This was a multicenter prospective study of 2915 women with a singleton pregnancy who underwent vaginal ultrasonography at approximately 24 and again at 28 weeks of gestation. Both women with and without a history of PTB were included. The association between cervical length and the risk of spontaneous PTB has been assessed. The main results of this study were: (1) At 24 weeks’ gestation, only 10% of women had CL < 26 mm; (2) The risk of PTB was inversely related to CL; (3) a CL < 26 mm at 24 weeks had a better predictive value compared to CL < 26 mm at 28 weeks (RR 6.1 vs. 5.3) for predicting PTB < 35 weeks. However, the positive predictive value (PPV) of a short CL, was low in this unselected population: Only 18% of women with a CL < 25 mm at 22–25 weeks of gestation delivered prior to 35 weeks of gestation [5].

In a different cohort of unselected women at 22–24 weeks of gestation, only 1.7% had a CL < 15 mm, but they accounted for 86% of all PTBs at <28 weeks of gestation and 58% of PTBs prior to 32 weeks of gestation [30]. It was suggested that the specificity was 99.9% for PTBs at <34 weeks of gestation for a CL cutoff of 20 mm. However, this value decreased to 90.1% and 65.5% for a CL threshold of 30 mm and 35 mm, respectively [31]. Salomon et al. [32] modeled cervical length in normal pregnancies and offered new reference values for cervical length based on a large sample. These centiles can be used to make a decision on the policy of treatments to reduce the morbidity and mortality of PTB. Currently, most major guidelines suggest using a mid-trimester CL threshold of 25 mm for risk assessment [10,11] (Table 1).

### 5.2. A Single vs. Repeated Measurements

Several studies have reported that the progressive shortening of transvaginal sonographic CL over time is associated with an increased risk of preterm birth [32,37,38,39], whereas others have not been able to demonstrate such an association [40]. In the study of Iams et al. [5], the change in CL between 24- and 28-weeks’ gestation was significantly associated with the risk of sPTB independently of the initial CL value. For women with cervix reduced in length throughout 24 and 28 weeks, the PTB rate was 4.2% compared to 2.1% of those whose cervix was relatively constant in length. The severity of the decline even had an impact on the likelihood of PTB; the relative risk was 2.80 (CI 1.87–4.20) for women whose cervixes had decreased by 6 mm or more compared to those whose cervixes had changed by less than 6 mm.

In an observational study of CL surveillance in 183 women with a history of one or more sPTBs [41], the predictive value of short cervix was examined in the early second trimester (16–19 weeks) and whether serial measurements had enhanced identification and prediction of sPTB for up to 24 weeks. The women who experienced an sPTB < 35 weeks shortened their cervixes at a median rate of 2.5 mm per week compared with a rate of 1.0 mm per week in the 130 women who did not (*p* = 0.03). The rate of change in CL (in terms of cervical slope) throughout the surveillance period was also shown to be an independent risk factor for sPTB, with women who had delivered prematurely showing a more rapid rate of cervical shortening. A steeper cervical slope was associate with the risk of sPTB even after controlling for a short CL at baseline.

In contrast, a recent systematic review [42] demonstrated that CL changes through time have been shown to have low prediction performance for PTB at <35 and <37 weeks of gestation in women with singleton gestation and weak to intermediate prediction performance for sPTB at <34, <32, <30 and <28 weeks of gestation in women with twin pregnancy. The authors concluded that changes in CL as a function of time cannot currently be considered a clinically useful test to predict an sPTB in women with singleton or twin gestations. They also considered that a single CL measurement taken at 18–24 weeks of pregnancy appears to be a better and more accurate test for predicting sPTB than changes in CL over time, and that it seems to be more cost-effective than serial CL measurements.

## 6. Who to Screen?—A Universal or Targeted CL Screen for PTB?

Universal CL screening can be used to identify asymptomatic women at risk of PTB, thereby providing an opportunity to offer interventions that may reduce that risk. Supporters of universal screening point to the findings of two major, randomized trials in which vaginal progesterone has been shown to reduce the risk of PTB in women with short cervix [16,17]. Even more support can be provided by two cost-effective analyzes, both showing universal cervical length screening as a cost-effective approach to reduce PTB [37,38].

The approach of CL screening varies according to the a priori risk of PTB. Currently, the Society for Maternal-Fetal Medicine (SMFM) and American College of Obstetricians and Gynecologists (ACOG) guidelines recommend that women with a prior sPTB will undergo CL screening with TVUS [10,39]. However, universal CL screening of singleton gestations without a prior sPTB for the prevention of sPTB remains a matter of debate [40,41]. A recently published meta-analysis of randomized controlled trials did not find sufficient evidence to recommend for or against routine CL screening of an unselected population, basically due to limitations in the methodology of the included trials [7]. Among the limitation of this meta-analysis were the limited number of included studies, a small sample size of the cohorts, and variation in the design and outcome measures of the different studies. The SMFM and ACOG acknowledge the debate surrounding this topic and state that although universal CL screening in women without a prior sPTB is not officially recommended, such a screening strategy may be considered [10,11] (Table 1). As CL assessment is debatable as a universal screening tool, more data has emerged, examining its value in predicting PTB in specific populations of women considered at high risk of preterm birth.

### 6.1. Women with Previous PTB

It is well established that a history of PTB or second-trimester miscarriage is one of the strongest predictors of PTB [9]. Previous studies assessed the value of a short CL in combination with an obstetrical history of sPTB [8,42,43,44]. Durnwald et al. [41] found similar rates of CL < 25 mm at 22–25 weeks’ gestation in women with a history of one previous sPTB (18–37 weeks of gestation) and in those with two or more sPTBs. Due to the limited sample size, it could not be determined whether an increasing number of prior sPTBs significantly modified the value of a short cervix in predicting a future sPTB prior to 32 or 35 weeks of gestation. Another study [43], with a larger sample size (*n* = 1958), aimed to estimate the probability of sPTB <34 weeks based on CL correlated with previous obstetrical history. They observed that the mean CL at 21–24 weeks’ gestation was 30.1 mm in the group with a previous history of PTB and 35.8 mm in the group without such history (*p* < 0.001). CL < 25 mm had a PPV of 12.5% in the latter group, but 33.5% in the former, emphasized the importance of the context of the obstetrical history for better risk stratification. Logistic regression analysis demonstrated that cervical length and a history of sPTB were independent contributors for preterm delivery. In contrast, we found that CL appears to be of limited value in predicting PTB among women with threatened preterm labor who are at high risk for preterm delivery owing to a history of sPTB in a previous pregnancy [44]. We postulated that the high a priori risk for PTB in this subgroup (i.e., symptomatic women with a history of PTB) may overweigh the predictive ability of CL for prediction of PTB, which was found to be relatively low beyond 24 weeks of gestation in symptomatic women [8,44].

A recent systematic review of clinical guidelines included 49 guidelines recommending clinical practices for the prevention or management of PTB in both asymptomatic and symptomatic women with risk factors for PTB. A global consensus (defined as agreement of >70% of the reviewed guidelines) for mid-trimester CL screening among women with a history of an sPTB was reached [45].

### 6.2. Women with a History of Treatment for Cervical Dysplasia

Treatment for cervical dysplasia is a recognized risk factor for PTB. A recent meta-analysis demonstrated that women with cervical intraepithelial neoplasia (CIN) have a higher baseline risk for prematurity. Excision and ablative treatment also increase this risk. The prevalence and severity of undesirable sequelae increases with increasing cone thickness and is greater for excision than for ablation [46]. Earlier, it was assumed that the extraction of tissue from the cervix undermined its structure and functions, which in turn may result in a shorter CL during pregnancy.

On the basis of that theory, previous studies have utilized cervical surveillance in women who have undergone surgery for cervical CIN. Pils et al. measured CL at fortnightly intervals from 16 to 22 weeks’ gestation in women with a previous history of one or more large loop excision in the transformation zone (LLETZ) in comparison to a control group of women who had a history of second-trimester miscarriage with no surgical manipulation of the cervix. The CL at 16 weeks of gestation was shorter in the LLETZ group (36 vs. 38 mm, *p* = 0.04) in the control group. The CL change rate was found to be very important in the LLETZ group, however the addition of serial CL measurements to the multivariate analysis revealed that subsequent measurements did not increase the predictive value of a single measurement at 16 weeks of gestation [47].

Berghella et al. [48] performed CL surveillance between 16- and 24-weeks’ gestation in 109 women after surgical treatment for CIN and analyzed the risk for PTB < 35 weeks’ gestation according to the type of previous cervical procedure. The predictive value of CL was calculated using a CL limit of less than 25 mm as the criteria for the classification of a short cervix. Sensitivity, specificity and positive and negative predictive values for sPTB were 64%, 78%, 30% and 94%. Cone biopsy, but not LLETZ, emerged as a significant risk factor for PTB. However, this study was limited by the fact that the significance of cervical slope and the optimal gestational age for predicting PTB was not assessed.

A recent study [49] aimed to investigate the value of serial CL measurement in predicting PTB among 613 post-conization women demonstrated that the maximal cervical length and the percentage change in CL measurement between 15–20 w are independent variables in predicting PTB.

There is currently substantial information to recommend additional CL screening for women with a prior electrosurgical procedure or a cold knife cone for cervical dysplasia [11].

### 6.3. Women with Uterine Anomalies

Although uterine malformation may increase the risk of PTB, the likelihood of sPTB appears to vary according to the type of uterine anomaly [50]. A small cohort (*n* = 64) study [51] indicated that a septate uterus possessed the best chance of a term delivery, and unicornuate the worst, with a PTB rate of 44%. While CL < 25 mm in the presence of some uterine malformation had a 50% PPV for delivery at <35 weeks in this specific study, that might not reflect the true association between the uterine malformation and sPTB.

Crane et al. [52] specifically examined the predictive value of CL for predicting PTB in women with uterine anomalies. They reported that women with a bicornuate or unicornuate uterus tended to have a shorter CL, and that shortening occurred earlier in women with uterus didelphys or uterine septum in comparison to low risk controls. In addition, it was noted that the use of CL cutoff <30 mm between 16 and 30 weeks of gestation had clinically unhelpful PPV (37.5%), but its 100% NPV would provide valuable reassurance for women undergoing surveillance and help avoid unnecessary prophylactic intervention. It is important to note that due to the extremely small sample of women with a septate or unicornuate uterus, generalizing the results to all subgroups of uterine anomalies is questionable.

A recent study [53] described outcomes of 319 women with uterine anomalies, in whom a short cervix <25 mm did not predict sPTB at <37 or <34 weeks. However, in subgroup analysis, a short cervix was shown to predict sPTB in women with resorption defects but not fusion defects.

Cervical surveillance appears to have a role in women with uterine malformation, however there is a paucity of data concerning this issue [51,52]. It lacks agreement on a significant cutoff for short CL, and it is currently unclear whether a short cervix carries a higher PPV for sPTB in women with uterine malformations than in other high-risk groups.

### 6.4. Multiple Pregnancy

It is well established that there is an increased risk of a PTB in women with multiple pregnancies and that cervical length is shorter than singleton pregnancies [54,55]. In a study that followed 1163 women with twin pregnancies until delivery, the median CL length at 22–24 weeks was 35 mm, and the rate of CL ≤25 mm, ≤20 mm, and ≤15 mm was 16%, 8%, and 5%, respectively. A CL of 10 mm, 20 mm, 25 mm, and 40 mm was correlated with a PTB < 32 weeks, which occurred in a rate of 66%, 24%, 12%, and <1%, respectively [56].

The association between a short cervix in asymptomatic women with multiple pregnancies and the risk of PTB was explored in 2 meta-analysis [57,58]. The less recent one evaluated 16 studies in which CL measurement was performed at 20–24 weeks of gestation [57].

Among asymptomatic women, a CL < 20 mm was the most accurate cutoff for predicting sPTB <32 and <34 weeks’ gestation (pooled sensitivities, specificities, and positive and negative likelihood ratios of 39% and 29%, 96% and 97%, 10.1 and 9.0, and 0.64 and 0.74, respectively). A CL < 25 had a pooled positive likelihood ratio of 9.6 to predict preterm birth <28 weeks’ gestation. In the second meta-analyses comprising 21 studies in which gestational age at screening ranged from 15 to 32 weeks, it was suggested that the sensitivity and specificity for PTB < 34 weeks were 36% and 94%, respectively, for CL < 25 mm, and 30% and 94% for CL < 20 mm [58].

Overall, among women with twin gestations, a CL of ≤20 mm at 20–24 weeks of gestation had pooled positive and negative likelihood ratios of 10.1 and 0.6, for predicting sPTB at <32 weeks of gestation, respectively, and 9.0 and 0.7, to predict sPTB at <34 weeks of gestation, respectively [59].

In asymptomatic women with twins and higher-order gestations, serial cervical length measurements was predictive of PTB [60,61,62,63], and integration of serial measurements of cervical length using a stepwise algorithm was shown to improve the detection of women at risk of preterm birth [61].

Melamed et al. [61] assessed the change in CL over time in 441 asymptomatic women with twin pregnancies who underwent TVUS to evaluate CL at timepoints between 18 and 32 weeks of gestation. The short cervix (<10th percentile) was linked with PTB at <32 weeks of gestation at any of the timepoints evaluated. The multistep process integrating repetitive CL measurements of all different time resulted in a significant improvement in the area under the receiver operating characteristic curve (0.917 vs. 0.613; *p* < 0.001). This group also found that the trend of cervical shortening was consistent with the prediction of PTB, and that rapid early shortening was correlated with the highest risk [62]. Those findings were supported by those of others [64].

Recently, a systematic review [7] concluded that due to the limited number of studies, including small numbers of women there are limited data on CL, measured by ultrasound, for preventing preterm births, which precluded from drawing any conclusions for women with asymptomatic twin pregnancy.

Several treatments are currently being investigated in randomized control trials for women with multiple gestation and shortened cervix, but at this point, the existing information do not suggest an appropriate therapeutic benefit to support routine checkups of all women with multiple gestation. Therefore, routine CL evaluation in multiple pregnancy is actually not recommended by the Society for Maternal-Fetal Medicine [39], although it is recommended by the Royal Australian and New Zealand College of Obstetricians and Gynecologists and by the International Society of Ultrasound in Obstetrics and Gynecology [33].

## 7. It Is Not All about Cervical Length

Although there have been major breakthroughs during the last decade, many questions remain unanswered regarding cervical remodeling and its role in risk assessment for sPTB. There has been growing interest in developing new techniques to assess changes in the cervix that could potentially help to improve the prediction of PTB in asymptomatic high-risk women.

### 7.1. Cervical Funneling

Cervical funneling is characterized as the dilation of the internal os of the cervix, with a bulge of the amniotic membrane into the endocervical canal. Although some studies used cervical funneling as an outcome (usually as a separate finding), cervical funneling does not significantly contribute to the risk of PTB correlated with shortened CL. A survey of Australian experts and sonographers reported that although CL measurement is a well-described technique with good interrater reliability, the consensus between evaluators is weak in the presence of funneling. This poses a strong obstacle to the evaluation of the use of cervical funneling as a predictor of sPTB [65]. In fact, while funneling is correlated with a short cervix, it is not an isolated indicator of sPTB itself once the enclosed size of the cervical canal is considered [66,67].

### 7.2. Amniotic Fluid Sludge

Sludge is intraamniotic debris usually lies near the internal os. It is a collection of particulate matter composed of biofilm and inflammatory cells [68], sourced either from ascending infection or from the placental microbiome, and indicating chronic intraamniotic infection. The presence of sludge on transvaginal sonography was found to be associated with a shorter CL, preterm premature rupture of membranes, and earlier gestational age at delivery [69].

Previous retrospective case-control study of 281 patients which aimed to assess the significance of sludge among asymptomatic women showed that wpmen with sludge had a higher probability of sPTB at <28 weeks (46.5 vs. 5.8%), <32 weeks (55.6% vs. 12.3%) and <35 weeks (62.2% vs. 19.9%). In addition, they found that the combination of a CL < 25 mm and sludge resulted an odds ratio of 14.8 and 9.9 for sPTB at <28 weeks and <32 weeks, respectively [70].

Another study [71] utilized to assess the association between intra-amniotic sludge and PTB has shown that the existence of amniotic fluid sludge among women at high risk for PTB who have been prescribed for second trimester CL measurements between 18 and 32 weeks of gestation is correlated with delivery within 14 days and delivery before 34 weeks of gestation. Delivery within 14 days of examination occurred in 5.3% of women with no sludge, 22.2% of women with light sludge, and in 60.0% of women with dense sludge. Delivery before 34 weeks occurred in 6.7%, 44.4%, and 80.0% of women, respectively. In addition, a recent study showed that amniotic fluid sludge is an independent risk factor for a PTB before 34 weeks of gestation in women with a CL <25 mm [72].

### 7.3. The Role of Cervical Elastography

To assess cervical elasticity is determined based on motion in areas of the cervix compared to other areas outlined in the color map. The cervix is examined in the conventional way. Gentle pressure is then applied with the transducer to modify the tissue, and advanced software is used to create a color map that illustrates the deformation of the tissue compared to the nearby areas. A case-control study [73] of 286 women who underwent TVUS in each trimester of pregnancy found that the mean strain value at the internal os of the cervix during the second and third-trimester can be helpful in predicting a PTB.

In a prospective cohort study [74], involving 628 women with a singleton pregnancy CL and cervical elastography of the internal os, were examined at 18–24 weeks of gestation. A soft cervix (identified by shear-wave elastography) at 18–24 weeks of gestation increases the risk of sPTB <37 and <34 weeks of gestation independently of cervical length. Moreover, in a recent meta-analysis [75] of seven studies that included 1488 pregnant women, cervical elastography showed a sensitivity of 0.84 and a specificity of 0.82. The authors concluded that cervical elastography is a promising and reliable method to predict a PTB.

### 7.4. Cervical Consistency

This method involves measuring the ratio of the anterior to the posterior diameter of the cervix before and after exerting pressure on the cervix. The reasoning behind such a strategy is that softer tissue changes more and has a lower cervical consistency index (CCI) than tougher tissue. To test this approach, 1031 women were seen once between 5 and 36 weeks of gestation, and the CCI decreased with increasing gestational age [76]. The sensitivity for predicting preterm birth by the CCI was 45–67%, compared to the 9–11% sensitivity of CL measurements [77]. Though the lack of an objective way to calculate the force used to manipulate the cervix is a major disadvantage of the CCI.

### 7.5. Acoustic Attenuation

Attenuation describes the concept of ultrasound signal amplitude decrease with distance from the transducer. Attenuation depends upon properties, such as hydration and microstructural organization. Since cervical remodeling involves extracellular matrix changes, such as increasing hydration and decreasing microstructural organization, measuring attenuation could potentially quantitate remodeling. In a previous report, 41 women (10–41 weeks of pregnancy) undergone TVUS, by which attenuation was measured. Attenuation was an indicator of the time from US evaluation to delivery, but not of gestational age or CL. Their study was not powered to determine attenuation as a predictor of a PTB, however, its potential was suggested by two cases of women with a short cervix, but who had moderate-to-high attenuation values at 18- or 29- weeks of gestation and both of them delivered at term. Those authors suggested that attenuation may be a better predictor of a PTB than CL [78]. A more recent study of 67 women that underwent TVUS examinations at 17–26 weeks of gestation reported that cervical attenuation was lower at 17–21 weeks of gestation in the sPTB group [79].

While the attenuation technique is promising, it will require thorough testing and verification of the correct interpretation of the findings before it is clinically viable.

### 7.6. Cervical Gland Area (CGA)

The absence of normal mucosal glands (cervical gland area), which is demonstrated by a hyperechoic or hypoechoic segment around the cervical mucosa, has also been proposed as a predictor of a PTB [80]. In past studies, it was shown that the detection rate of the CGA decreased after 31 weeks of gestation [81]. This absence of glands may signify the beginning of the cervical ripening process [80,81,82]. In another study, 600 pregnant women were scanned at 16–19 weeks of gestation, and the CGA was detected in 77% of the women who delivered at term vs. 55% of those who did not [80].

### 7.7. Fetal Fibronectin (fFN)

Another marker that has been incorporated widely as a test for PTB among asymptomatic women is fFN, a glycoprotein released from the interface of the decidua and chorion during preclinical labor. It was originally used it as a qualitative test wherein the results of 50 ng/mL or more are considered positive, the main benefit being its high negative predictive value [83], although this is related to low prevalence and sensitivity of <90%.

Quantitative fFN is the measure of absolute concentration of fFN, and prediction has been improved relative to qualitative fFN in a cohort of asymptomatic women [84]. The assessment of the quantitative instrument used to forecast PTB was the first prospective research to show the enhanced importance of quantitative fFN analysis. The use of multiple fetal fibronectin thresholds (10, 50, 200, and 500 ng/mL) increased the PPV value for PTB in asymptomatic high-risk women [84]. Two studies compared and combined fFN with CL as a predictor of a PTB in asymptomatic women, and both found fFN to be a superior test [85,86,87]. However, others have demonstrated that fFN results alone are not useful [88,89,90]. In a systematic review [91] of six randomized trials in which a total of 546 women with threatened preterm labor were randomly assigned to management with reported or concealed fFN results, clinician knowledge of their fFN results did not reduce the rates of maternal hospitalization or a PTB at <34 weeks of gestation. Moreover, a recent prospective observational cohort study of nulliparous women with singleton pregnancies demonstrated that quantitative vaginal fFN and serial TVUS-determined CL had low predictive accuracy for an sPTB [29].

## 8. Shortcomings of Current Data

Studies evaluating cervical characteristics and risk of PTB are extremely heterogeneous with respect to the target population, the treatment obtained, and the outcomes of concern. The populations of the studies examined were classified as high risk of sPTB, but most were diverse because the inclusion criteria differed greatly from one study to the other. Conditions known to increase the risk of PTB have been omitted from certain studies, but not others. Studies have not often focused on the inclusion of women with these risk factors, making comparisons between experiments challenging. Alternatively, in some situations, researchers would appear to concentrate on CL surveillance in the presence of a single risk factor for PTB, but would also involve women with multiple risk factors. Another issue that complicates comparisons is that the classification of preterm births differs from study to study. Some also include only slightly and severely premature births as their primary outcomes, while others take a broader viewpoint by using 37 weeks of gestation as a cutoff point.

It is also plausible that specific pathophysiology underlies severe versus late PTBs, and that a single evaluation or treatment method might not be applicable to several potential mechanisms. Recent studies have attempted to classify sPTBs according to various phenotypes, which may provide a valuable framework for reporting further studies on sPTB predictors [92].

## 9. Conclusions

The magnitude of the problem of prematurity and the seriousness of its effects have given rise to comprehensive research into the possible causes and predictors of PTB and preventive strategies, although many issues remain unsolved. The challenge in researching a single outcome is that different mechanisms can well rely on different initiators and risk factors. In addition to this issue, the current research body involves a heterogeneous population with a diversity of risk factors for sPTBs, certain of which have not been studied separately. It may explain why CL is continuously emerging as an important yet disappointingly imprecise predictor of sPTB in high-risk women. While it may be suggested that this reflects the real-world application of the test, it seems unwise to assume that a single variable will act as a universal predictor when various mechanisms are involved. Assuming that PTB is a consequence of various, complex pathways, a variety of methods is likely to be needed for a thorough risk assessment. In addition, clinical and obstetric parameters are likely to continue to be integrated into novel imaging techniques, with the possibility that new phenomena may be discovered that are significant in the process of human cervical remodeling.

## Figures and Tables

**Table 1 jcm-10-00627-t001:** Summary of major guidelines regarding the different aspects of CL screening and technique.

Organization	Recommendation	Grade
Society for fetal maternal medicine (SMFM) [11]	Routine transvaginal CL screening for women with a singleton pregnancy and history of prior spontaneous PTB at 16–24 weeks’ gestation.	A
	Routine transvaginal CL screening to not be performed for women with cervical cerclage, multiple gestations, PPROM, or placenta previa.	B
	Routine CL screening in multiple pregnancies is not currently recommended.	B
American College of Obstetricians and Gynecologists (ACOG) [10]	Routine transvaginal CL screening for women with a singleton pregnancy and history of prior spontaneous PTB starting 16–24 weeks’ gestation.	A
	Although the ACOG does not mandate universal cervical length screening in women without a prior preterm birth, this screening strategy may be considered.	B
International Society of Ultrasound in Obstetrics and Gynecology (ISOUG)	For twin pregnancies, cervical length measurement is the preferred method of screening for preterm birth in twins; 25 mm is the cutoff most commonly used in the second-trimester [33].	A
	Currently, there is insufficient evidence to recommend routine cervical length measurements at the mid-trimester in an unselected population [34].	B
Society of Obstetricians and Gynecologists of Canada (SOGC) [35,36]	Transvaginal ultrasonography is the preferred route for cervical assessment to identify women at increased risk of spontaneous preterm birth and may be offered to women at increased risk of preterm birth.	B
	Because of poor positive predictive values and sensitivities and lack of proven effective interventions, routine transvaginal cervical length assessment is not recommended in women at low risk.	B
	There is no consensus on the optimal timing or frequency of serial evaluations of cervical length. If repeat measurements are performed, they should be done at suitable intervals to minimize the likelihood of observation error.	B
	There are insufficient data to recommend a routine preterm labor surveillance protocol in terms of frequency, timing, and optimal cervical length thresholds for twins’ pregnancies.	B

CL—cervical length, PTB—preterm birth, PPROM—premature pre-labor rupture of membranes. Grade A—strong recommendation, high-quality evidence. Grade B—weak recommendation, moderate-quality evidence.

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
