# Peer review of "Cervical Assessment for Predicting Preterm Birth—Cervical Length and Beyond"

_jcm, 2021, doi:10.3390/jcm10040627_

Round 1

Reviewer 1 Report

Manuscript number jcm-1054596

Reicher L and colleagues have performed a well-written review of Cervical Assessment for Predicting Preterm Birth.

Comments;

The paper is well structured – and the headings/sections are in relevant order. In each section (1-6), the authors managed to review shortly and precise ‘up-to-date’ knowledge with relevant references.

Further, PROS and CONS for universal CL screening for PTB are discussed as well as CL screening for more specific high-risk group.

The pathophysiology of PTB is very complex. These multifactorial risk factors for PTB and the challenges of comparing study population due to high heterogenicity is well summarized and discussed conclusion of the review.  If possible, these two sections could be a bit shorten so it is a bit more readable.  

Introduction;

Linie 22-25; this could maybe be rewritten so the sentences will be shorter and more easily to read.

The sections; 7. It is Not All about Cervical Length;

I would recommend that the length of the sections  ‘Funneling’, ‘Sludge’ as well as ‘fFN’ will be reduced. Especially, funneling and fFN which do not seem to be independent risk factors for PTB.

Linie 93; 'unelected' correct to unselected

Author Response

Response to Reviewer 1 Comments

Point 1: Line 22-25; this could maybe be rewritten so the sentences will be shorter and more easily to read.

Response 1: The paragraph has been revised

Point 2: The sections; 7. It is Not All about Cervical Length;

I would recommend that the length of the sections ‘Funneling’, ‘Sludge’ as well as ‘fFN’ will be reduced. Especially, funneling and fFN which do not seem to be independent risk factors for PTB.

Response 2: These sections have been shortened

Point 3: Line 93; 'unelected' correct to unselected

Response 3: Corrected

Reviewer 2 Report

This is a nicely written summary of a surprisingly complex topic. I like the way the sections are arranged by question, and the table clearly summarises the agreements, and disparities, of recommendations from international groups. I think this paper will be useful for clinicians considering introducing CL screening/wanting to find out more about it.

More specific points (with doi for references):

Line 73: I think we normally say 'gold standard' not 'golden standard'

Line 75: I think I'm being very pedantic, but it can be harder to obtain a CL in high BMI, and it is harder if there is a cervical pessary that is affecting cervical position, suggest change to less influenced by maternal obesity...

Section 5.1: The authors allude to the expected slight shortening of CL with advancing gestational age (and so CL <26mm at 24 weeks gestation has a better predictive value for PTB than at 26 weeks gestation). This discussion might be improved by reference to Salomon et al (DOI: 10.1002/uog.6332) who modelled cervical length in normal pregnancies and showed this decline quite clearly. It had been suggested to initiate PTB prevention treatment based on a centile of these charts, but I don’t think that has widely been adopted.

Section 5.2: I am not sure why you have explicitly described the Owen study (reference 37), and then refuted this with the larger systematic review of change in CL over time (Conde-Agudelo and Romero, reference 38), which did include the Owen study. I would judge the systematic review to be more robust and the section discussing the Owen study could be removed- unless there is something else especially pertinent about the Owen study that I have not understood- if so could that be more clearly explained please?

Introduction to section 6: I think it would be helpful this section highlight the main reason for CL scanning in asymptomatic women- to target PTB prevention treatment

Section 6.1: I feel this section is slightly muddled by combing, and then not combing, women with and without symptoms of preterm labour. The final sentence (line 169) about the global consensus for mid trimester CL screening for women with previous sPTB feels like the strongest point in this section. I would start with that, discuss evidence around it, and then in at least a separate paragraph, maybe a separate section, discuss symptomatic women. I would also recommend reference to the work of Shennan et al for predictive modelling of symptomatic and asymptomatic women using CL (eg doi: 10.1002/uog.20401 and https://doi.org/10.1002/uog.20422).

In the UK NICE (https://www.nice.org.uk/guidance/ng25) offer the option of progesterone for PTB prevention in the case of previous sPTB<34 weeks gestation regardless of CL. You may want to consider the argument for no CL screening and treatment regardless of CL in this high risk/high morbidity group.

Section 6.2: You quote the Kyrgiou review, 2014 (ref 46) and the Conner review (ref 50) into CL screening after cervical surgery. I am not sure I agree with the statement that the elevtated risk of sPTB is related to the history of cervical dysplasia rather than the procedure itself, because the Kygiou review was updated in 2016 (doi: https://doi.org/10.1136/bmj.i3633), and clearly states the opposite view. Also, this section currently gives us an overview of CL after cervical surgery, but isn’t the more pertinent question whether CL screening in women who have had previous cervical surgery helps to prevent PTB? Could the focus of this section be altered to cover the more clinically important question?

Line 210: I think this has an error where “[earlier in comparison to?].” is stated- I agree earlier in comparison to whom would be helpful

Section 6.4: You have comprehensively explained that short cervical length has a weak association with PTB in twins. The common reason given for not performing CL scanning in twins is that no PTB prevention treatment has been shown to prevent PTB in this twins (maybe pessary if cervix very short DOI:https://doi.org/10.1016/j.ajog.2017.11.598), I think this should be highlighted.

Lines 304-307: I think this paragraph would benefit from risk/percentages to contextualise the statement of ‘is associated with’

Lines 332-333: You quote the diagnostic accuracy of cervical elastography in comparison to CL measurement from a preliminary report from 2010 (ref 78) instead of the earlier mentioned 2019 systematic review and metanalysis (reference 76). I think focusing on reference 76 may be more appropriate.

Section 7.7- this seems to focus only on asymptomatic women (as does most of section 7). It may be clearer to state that at the beginning, especially as fFN appears to perform better in symptomatic women, so your conclusions are clearer if we all understand we are in the asymptomatic situation. Another prospective observational study that found no additional value of fFN in addition to CL and a treatment algorithm was https://doi.org/10.1111/1471-0528.15886.

Reference 86- this group have gone on to do further work about fibronectin and CL, I think a more recent reference would be appropriate (eg doi: 10.1002/uog.20401 and https://doi.org/10.1002/uog.20422).

Table 1: I may be very UK centric, but NICE have also developed comprehensive PTB guidelines that may warrant inclusion in your table ((https://www.nice.org.uk/guidance/ng25). This table is a very clear summary and really adds to the paper.

Author Response

Response to Reviewer 2 Comments

Point 1: Line 73: I think we normally say 'gold standard' not 'golden standard'

Response 1: The sentence has been corrected

Point 2: Line 75: I think I'm being very pedantic, but it can be harder to obtain a CL in high BMI, and it is harder if there is a cervical pessary that is affecting cervical position, suggest change to less influenced by maternal obesity...

Response 2: The sentence has been corrected

Point 3: Section 5.1: The authors allude to the expected slight shortening of CL with advancing gestational age (and so CL <26mm at 24 weeks gestation has a better predictive value for PTB than at 26 weeks gestation). This discussion might be improved by reference to Salomon et al (DOI: 10.1002/uog.6332) who modelled cervical length in normal pregnancies and showed this decline quite clearly. It had been suggested to initiate PTB prevention treatment based on a centile of these charts, but I don’t think that has widely been adopted.

Response 3: We have added a paragraph describing Solomon's paper to this section.

Point 4: Section 5.2: I am not sure why you have explicitly described the Owen study (reference 37), and then refuted this with the larger systematic review of change in CL over time (Conde-Agudelo and Romero, reference 38), which did include the Owen study. I would judge the systematic review to be more robust and the section discussing the Owen study could be removed- unless there is something else especially pertinent about the Owen study that I have not understood- if so could that be more clearly explained please?

Response 4: The paragraph describing Owen's study has been removed.

Point 5: Introduction to section 6: I think it would be helpful this section highlight the main reason for CL scanning in asymptomatic women- to target PTB prevention treatment

Response 5: We've added an introduction to this section.

Point 6: Section 6.1: I feel this section is slightly muddled by combing, and then not combing, women with and without symptoms of preterm labour. The final sentence (line 169) about the global consensus for mid trimester CL screening for women with previous sPTB feels like the strongest point in this section. I would start with that, discuss evidence around it, and then in at least a separate paragraph, maybe a separate section, discuss symptomatic women. I would also recommend reference to the work of Shennan et al for predictive modelling of symptomatic and asymptomatic women using CL (eg doi: 10.1002/uog.20401 and https://doi.org/10.1002/uog.20422).

Response 6: The paragraph has been edited. This paper focuses solely on asymptomatic women.

Point 7: Section 6.2: You quote the Kyrgiou review, 2014 (ref 46) and the Conner review (ref 50) into CL screening after cervical surgery. I am not sure I agree with the statement that the elevtated risk of sPTB is related to the history of cervical dysplasia rather than the procedure itself, because the Kygiou review was updated in 2016 (doi: https://doi.org/10.1136/bmj.i3633), and clearly states the opposite view. Also, this section currently gives us an overview of CL after cervical surgery, but isn’t the more pertinent question whether CL screening in women who have had previous cervical surgery helps to prevent PTB? Could the focus of this section be altered to cover the more clinically important question?

Response 7: The paragraph has been revised and now contains three studies on the predictive value of CL screening for PTB among women who had had previous cervical surgery.

Point 8: Line 210: I think this has an error where “[earlier in comparison to?].” is stated- I agree earlier in comparison to whom would be helpful

Response 8: The mistake was corrected.

Point 9: Section 6.4: You have comprehensively explained that short cervical length has a weak association with PTB in twins. The common reason given for not performing CL scanning in twins is that no PTB prevention treatment has been shown to prevent PTB in this twins (maybe pessary if cervix very short DOI:https://doi.org/10.1016/j.ajog.2017.11.598), I think this should be highlighted.

Response 9: A paragraph has been added to this section. "Various interventions are currently being tested in RCT’s for women with multiple gestation and shortened cervix, but at this time available data does not indicate adequate clinical benefit to justify routine screening of all women with multiple gestations."

Point 10: Lines 304-307: I think this paragraph would benefit from risk/percentages to contextualise the statement of ‘is associated with’

Response 10: These data has been added to the paragraph.

Point 11: Lines 332-333: You quote the diagnostic accuracy of cervical elastography in comparison to CL measurement from a preliminary report from 2010 (ref 78) instead of the earlier mentioned 2019 systematic review and metanalysis (reference 76). I think focusing on reference 76 may be more appropriate.

Response 11: The systematic review refers to the elastography and the preliminary report refers to cervical consistency.

Point 12: Section 7.7- this seems to focus only on asymptomatic women (as does most of section 7). It may be clearer to state that at the beginning, especially as fFN appears to perform better in symptomatic women, so your conclusions are clearer if we all understand we are in the asymptomatic situation. Another prospective observational study that found no additional value of fFN in addition to CL and a treatment algorithm was https://doi.org/10.1111/1471-0528.15886.

Response 12: We've added a statement "among asymptomatic women" and added the mentioned ref.

Point 13: Reference 86- this group have gone on to do further work about fibronectin and CL, I think a more recent reference would be appropriate (eg doi: 10.1002/uog.20401 and https://doi.org/10.1002/uog.20422).

Response 13: We've added this ref.

Point 14: Table 1: I may be very UK centric, but NICE have also developed comprehensive PTB guidelines that may warrant inclusion in your table ((https://www.nice.org.uk/guidance/ng25). This table is a very clear summary and really adds to the paper.

Response 14: Those guidelines refers to the management of PTB and doesn’t provide any recommendations for CL surveillance.

This manuscript is a resubmission of an earlier submission. The following is a list of the peer review reports and author responses from that submission.